# Effect of the Application of Hydrolysate of *Chlorella vulgaris* Extracted by Different Techniques on the Growth of *Pelargonium* × *hortorum*

**DOI:** 10.3390/plants11172308

**Published:** 2022-09-03

**Authors:** Pedro José Bayona-Morcillo, Cintia Gómez-Serrano, Cynthia Victoria González-López, Daniele Massa, Silvia Jiménez-Becker

**Affiliations:** 1Department of Agronomy, Higher Engineering School, Agrifood Campus of International Excellence (CeiA3), Ctra. Sacramento s/n., 04120 Almería, Spain; 2Department of Engineering, Higher Engineering School, Agrifood Campus of International Excellence (CeiA3), Ctra. Sacramento s/n., 04120 Almería, Spain; 3CREA Research Centre for Vegetable and Ornamental Crops, Council for Agricultural Research and Economics, Via dei Fiori 8, 51017 Pescia, Italy

**Keywords:** biostimulants, ultrasound, enzymatic hydrolysis, autoclaving, ornamental plants

## Abstract

The extraction method used to obtain biologically active compounds from microalgal biomass may affect the biostimulant capacity of the microalgae. The objective of this assay was to determine the most efficient extraction method to release the active components of the biomass of *Chlorella vulgaris* (*C. vulgaris*). Plantlets of *Pelargonium* × *hortorum* were grown in a greenhouse and five treatments were applied: C-application with water; M-application with untreated *C. vulgaris* microalgae; M-US-application with *C. vulgaris* microalgae treated with ultrasound; M-USHY-application with *C. vulgaris* microalgae treated with ultrasound and enzymatic hydrolysis; and M-USHYAU-application with *C. vulgaris* microalgae treated with ultrasound, enzymatic hydrolysis, and autoclaving. All microalgae treatments increased shoot number and stem and plant diameter. The US-treated biomass increased the inflorescence of the plant significantly compared to the control. To extract bioactive compounds from eukaryotic microalgae for plant biostimulating purposes, the US-treatment (or any other method damaging the plasma membrane) of microalgae cell is, or seems to be, suitable.. Macronutrient content in leaves was not affected by the microalgae treatment, except for K.

## 1. Introduction

A plant biostimulant is any substance or microorganism applied to plants with the aim to enhance nutrition efficiency, abiotic stress tolerance and/or crop quality traits, regardless of its nutrients [1]. Biostimulants are products derived from organic material that, when applied in small quantities, are able to stimulate the growth and development of several crops under both optimal and stressful conditions [1]. Biostimulants belong to three main groups: humic substances, algal extracts, and products containing amino acids [2]. Microalgae are well-known because of their application as bioremediation players, due to their outstanding capacity to sequester carbon dioxide from the atmosphere, and as raw materials thanks to the wide variety of bioactive compounds that they are made up of [3]. Furthermore, microalgae have a high growth rate, photosynthetic efficiency, and a strong adaptability to the environment [4]. Microalgae are unicellular microorganisms which thrive in saltwater, freshwater environments [5], and wastewater, thus allowing a reduction in production costs [6]. Microalgal species can use nutrient from wastewater and dwell efficiently. Therefore usage of high-cost raw materials or chemicals as nutrients can be easily replaced, which will reduce the cost of production [7]. In particular, *Chlorella* sp. exhibits the potential to treat wastewater [7]. Microalgae are recognized as rich raw materials since they are composed of a large plethora of bioactive compounds, namely pigments (such as carotenoids and chlorophylls), proteins, carbohydrates, lipids, vitamins, phenolic compounds, and antioxidant compounds, all of which have been applied over the years in different industries, including cosmetics, pharmaceuticals, healthcare industries, animal feed, human food, bioenergy, and agriculture [3,4,8,9]. Furthermore, the nitrogen content of microalgae serves as a preferable organic fertilizer to ultimately reduce the reliance on chemical fertilizers, which pose an environmental hazard, particularly to soil and nitrogen [4]. In addition, the C:N:P ratio of microalgae makes them a potential feedstock for the various sectors such as bioenergy, bioremediation, biofertilizer, pharmaceuticals, and immunology [4].

Since most of the compounds are locked within the rigid cell walls, the extraction of bioactive compounds from eukaryotic microalgae involves cellular disruption to release the intracellular content so that it can be accessed and purified [5]. The extraction processes can be achieved by: (i) mechanical/physical means (e.g., autoclaving, homogenisation, microwaving, pulsed electric field technology, and sonication, liquid nitrogen); (ii) chemical methods (e.g., sodium hydroxide, hydrochloric or sulfuric acids, osmotic shock, nitrous acid), and (iii) enzymatic means (e.g., cellulase, protease) [10]. Recently, scientists have turned their attention to novel methods, such as enzyme-assisted extraction, microwave-assisted extraction, pressurized liquid extraction, supercritical fluid extraction, and ultrasound-assisted extraction, which enable the extraction of biologically active compounds without their degradation [11].

The dominant species of microalgae in commercial production include *Isochrysis*, *Chaetoceros, Chlorella, Arthrospira (Spirulina),* and *Dunaliella* [12]. *C. vulgaris’* appeal is its minimal nutrient requirements since they are unicellular, photosynthetic, and fast-growing microorganisms [13]. The composition of *C. vulgaris* varies depending on the nutrient supply and other environmental conditions, but under ideal conditions (30 °C, pH between 8.5 and 11, and 12:12 light/dark cycle of light source), *C. vulgaris* has been found to contain 45.23% protein, 23.43% carbohydrate, and 18.12% total lipids [14], as well as having a higher number of water-soluble vitamins compared to other microalgae [14]. Additionally, this microalga has brassinosteroids and gibberellin hormones [15], along with high concentrations of glutamine (9.38 g to 11.71 g Glx per 16 g N) [16]. Prabakaran et al. [14] found that *C. vulgaris* grown in mass cultures under optimal conditions (30 °C, pH between 8.5 and 11 and 12:12 light/dark cycle of light source) contains 4.7 mg/g, 4.2 mg/g, and 6.11 mg/g of chlorophyll a, b pigments, and carotenoids, respectively [14], and the predominant fatty acids are C18:3 n-3, C18:2 n-6 cis, and C16:0 [16]. Phenolic compounds were also identified in *C. vulgaris* [4]. *C. vulgaris* is a source of high-value bioactive compounds. An efficient extraction technique is necessary to obtain the maximal bioactive compound content from microalgae. Recent efforts to extract and purify some components from microalgae have increased, but more systematic studies are required to identify cost-effective methods to produce maximum bioactive compounds from marine sources [7,9]. Additionally, the bioactive compounds of the microalgae can exert synergistic effects; therefore in vivo studies are needed to establish the most appropriate extraction method [4].

The objective of this assay is to determine the most efficient extraction method to release the active components of the biomass of *C.vulgaris*. We conducted a comparative study of the growth promoting effects on *Pelargonium* × *hortorum* of the biomass of *C.vulgaris* treated with different extraction methods (ultrasound, enzymatic hydrolysis, and autoclaving).

## 2. Results

All *C. vulgaris* treatments increased stem diameter, shoot number, and plant diameter compared to the control (Figure 1a,b and Figure 2a). All the microalgae treatments tested had stem diameters ranging from 0.81 to 0.89 cm. As far as the number of shoots per plant was concerned, this increased when the plants were treated with microalgae (M, M-US, M-USHY, and M-USHYAU), being 2.8, 2.9, 2.9, and 2.6 shoots per plant, respectively, showing no significant differences among them. Plant diameters were significantly greater when the plants were sprayed with microalgae M (20.39 cm), M-US (21.53 cm), M-USHY (20.65 cm), and M-USHYAU (20.83 cm), showing significant differences with respect to the control (18.55 cm).

Differences in number of leaves and inflorescence were found (Figure 2b and Figure 3a). Plants sprayed with the biomass of *C. vulgaris* treated with ultrasound, enzymatic hydrolysis, and autoclaving resulted in an increase of more than 23% and 60% in the number of leaves and inflorescence compared to the control. Plants sprayed with biomass treated with ultrasound resulted in a 56% increase in the inflorescence number compared to the control. The rest of the treatments with microalgae showed intermediate values.

The ultrasound treatment, enzymatic hydrolysis, and autoclaving treatment, advanced flowering compared to the other treatments. The M-USHYAU treatment and the ultrasound treatment (M-US) achieved 100% inflorescence from day 40 of the experiment, compared to only 30% with the control treatment (Figure 3b).

The best results for leaf area were obtained by the ultrasound, enzymatic hydrolysis, and autoclaving treatments (Figure 4a), being approximately 27% greater with respect to the control, and with no significant differences among the other microalgae treatments.

Regarding dry weight, there were differences between treatments (Figure 4b). Microalgae application was effective in inducing leaf and flower and stem and petiole dry weight in treatments M and M-USHYAU, with no differences with the M-US treatment. The leaf and flower dry weights for the C, M, M-US, M-USHY, and M-USHYAU treatments were 0.88, 1.13, 1.09, 0.95, and 1.19 g, respectively. The stem and petiole dry weights for the C, M, M-US, M-USHY, and M-USHYAU treatments were 0.42, 0.54, 0.47, 0.39, and 0.52 g, respectively. No differences between M-US, M-USHY, and C in stem and petiole dry weight were found. Root dry weight was not affected by the treatments.

Water content, root dry weight, plant height, and red and green colour were not affected by the microalgae treatments (Figure 5a,b and Figure 6). The greatest amount of blue was found in C and M-USHYAU, followed by M-UHY and M. M-US.

Nitrogen, K, Ca, Mg, and S content in leaves was not affected by the microalgae treatments (Figure 7a,b). However, P decreased in the treatments with ultrasound, enzymatic hydrolysis, and autoclaving. No differences in P content were found between C, M, M-US, and M-USHY.

## 3. Material and Methods

### 3.1. Microalgae Biomass Production

The microalga *C. vulgaris* was grown in a bubble column with a volumetric capacity of 80 L. The culture medium was made from fertilizers which are used for agricultural purposes: 0.9 g L^−1^ of NaNO_3_, 0.18 g L^−1^ of MgSO_4_, 0.14 g L^−1^ of KH_2_PO_4_, and 0.02 g L^−1^ of Karentol mix. This culture was kept in a batch until the stationary phase had been achieved, and it was then collected by centrifugation. The original *C. vulgaris* culture, whose concentration was 2 g/L, was centrifuged with an alfa-laval centrifuge which concentrated the biomass to 200 g/L (20%DW), and it was then diluted to the desirable concentration of 100 g/L (10%DW).

### 3.2. Microalgae Application and Treatments

The microalgae were applied to the plant by foliar spraying as well as direct application to the substrate every 15 days (0, 15, 30, and 45 days after transplanting). The microalgal concentration was 0.1 g L^−1^ (dry weight). The foliar application was carried out via 10 sprays, and 50 mL was applied directly to the substrate for each treatment. There were 12 replicates per treatment.

The treatments tested were: C-application with water; M-application with untreated *C*. *vulgaris* biomass; M-US-application with *C*. *vulgaris* microalgae treated with ultrasound; M-USHY-application with *C. vulgaris* microalgae treated with ultrasound and enzymatic hydrolysis; and M-USHYA-application with *C. vulgaris* microalgae treated with ultrasound, enzymatic hydrolysis, and autoclaving.

The protein, lipid, and carbohydrate content in untreated *C. vulgaris* were 42.68 ± 1.07, 3.80 ± 0.40, and 46.48 ± 0.80% respectively. The hormone content of *C. vulgaris* are presented in Table 1.

To achieve the desired products for the application of the treatments, the biomass at 100 g/L was processed as follows:

Sonication (UP400S, Hielscher) was performed for 5 min using an amplitude of 100% and cycle 1, with the aim of causing a rupture of the cell wall to release the intracellular contents.

Enzymatic hydrolysis of the ruptured biomass was carried out using jacketed reactors, stirred via an impeller; the temperature was 40 °C and the pH was 8, which was established by the addition of sodium hydroxide (NaOH) 1N. For hydrolysis, the enzymes Alcalase 2.5 L and Flavourzime 1000 L (Novozyme) were used at a dose of 7% *v/w* of protein content. The procedure took 4 h.

Autoclaving of the hydrolysed biomass was achieved using a J. P. Selecta Mod. 4004371 (Dicsa, Almeria, Spain) autoclave for 15 min at a temperature of 121 °C.

### 3.3. Greenhouse Trial

The trials were conducted in a 170 m^2^ greenhouse equipped with zenith ventilation, relative humidity, and temperature control (University of Almeria (36°490′ N, 2°240′ W)). The trial was conducted between March and April 2021. The relative humidity and temperature were recorded every 15 min with a HOBO U12-013 data logger placed at canopy height in the central part of the cultivation table where the plants were grown. To estimate the internal radiation, the cover transmission coefficient was estimated as a ratio between the internal and external radiation using a manual quantum photo-radiometer (Detal OHM, model RAD/PAR) on clear days at 12:00 (Greenwich Mean Time). External radiation was obtained from the weather station located in Almeria, belonging to the Junta de Andalucía (36°50′07″ N 02°24′08″ W). Internal radiation was estimated as the multiplication of external radiation by the cover transmission coefficient. The average temperature, humidity, and global radiation were 17.12 °C, 69.95%, and 1.47 MJ m^2^ day^−1^, respectively; the maximum and minimum temperature and humidity averages were 23.62 °C and 12.66 °C, respectively, and 79.54% and 54.79%.

*Pelargonium* × *hortorum* var. ‘Dolce Vita DM Rose Eye (Claudio)’ were transplanted in 1.5 L volume pots, using a mixture of peat and perlite 70:30 (*v/v*). Fertigation was applied manually until the leachate fraction reached 20%. The average dose for *Pelargonium* × *hortorum* was 55 mL per plant per day. The concentrations in the standard nutrient solution were: 1.52, 11.50, 2.25, 6.00, 3.75, and 1.75 mmol L^−1^ of H_2_PO_4_^−^, NO_3_^−^, SO_4_^2−^, K^+^, Ca^2+^, and Mg^2+^, respectively. The pH and EC for the control treatment were 6.3 and 2.61 dS m^−1^, respectively.

### 3.4. Biometric Parameters

Sampling was carried out at the end of the assay, 53 days after transplanting. Plant height, stem diameter, leaf length and width, as well as plant diameter, were measured using a flexible handheld ruler and a vernier caliper. Plant height was determined from the base to the apex tip of the plant. Plant diameter was calculated as the average value of two diameters, measured perpendicularly. The number of leaves and shoots per plant were counted at the end of the crop, and the inflorescence was counted during the crop cycle every 4–5 days. After removing the substrate, the plant material was separated into different organs: roots, stems and petioles, and leaves and flowers, which were weighed separately on a COBOS G M5-1000 scale (with a precision of 0.005 g) to determine the fresh weight (FW). All the samples were washed and dried in a Nüve EFN500 oven (range from 30 to 300 °C) at 60 °C for 48 h, and the dry weight was recorded (DW). The total DW was calculated as the sum of the roots, stems and petioles, and leaves and flowers. The total FW and DW were used to calculate the water content (WC) as (FW-DW)/FW. The leaf area was estimated by a non-destructive method, using the formula S = a + bLW, as proposed by [17], where S is the foliar surface area, L is the leaf length (cm), W is the leaf width, and the a (0.07) and b (0.68) coefficients are specific to each species. For leaf colour determination, first, a scanner, Brother Model MFC-J4710DW, was used to scan 4 leaves for each treatment. Then, the red, green, and blue values (0–255) of the leaves were abstracted from the scanned picture by Image J software 1.52p, creator Wayne Rasband, Wisconsin, USA.

### 3.5. Plant Analysis

A subsample of the dry matter of the leaves was ground up in a Wiley mill and digested (in 96% H_2_SO_4_) to analyse the organic N, P, K, Ca, and Mg. The total K^+^ was directly measured by flame spectrophotometry using an Evans Electro Selenium LTB Flame Photometer (Halstead, Essex, England). Total Ca^2+^ and Mg^2+^ were analysed by atomic-absorption spectrophotometry using a Perkin Elmer Atomic Absorption Spectrometer 3300 (Dicsa, Almería, Spain). Phosphorus was analysed using the method proposed by [18] and N was analysed using the method proposed by [19].

### 3.6. Experimental Design and Statistical Analysis

The experimental design was comprised of a completely randomised block with 5 treatments. There were 12 replicates per treatment. A one-way Analysis of Variance (ANOVA) and a Least Significant Difference (LSD) test (*p* < 0.05) were performed using Statgraphics Centurion 18 (Statpoint Technologies, Inc., Warrenton, VA, USA), in order to evaluate the differences between treatments; these are represented by lowercase letters.

## 4. Discussion

Microalgae are known to contain vast quantities of high-value phytochemicals [4]. A study conducted by Gitau et al. [20], reported that *C.*
*vulgaris* application led to more robust plants compared to the control. According to our previous results, microalgae application stimulated lateral shoot growth [21]. When compared to the control, the number of shoots increased by 27 to 34% [21], when *Arthrospira* were applied. Similarly increase as this assay. In this assay stem and plant diameter also were positively affected by the application of *C.*
*vulgaris*. The phytohormones present in this microalga may be involved in that improvement. Endogenous cytokinins and auxins have been detected in *C. vulgaris* [15]. *C. vulgaris* has a higher concentration of isopentenyl adenine (Table 1), which can accelerate plant development. The application of cytokinin to *Rhododendron indicum* stimulates shoot regeneration [22]. *C. vulgaris* is rich in carbohydrates and proteins [23]. Carbohydrates play an important role in the life of plants: they are structural and storage substances, respiratory substrates, and intermediate metabolites of many biochemical processes [24]. The application of microalgae polysaccharides showed an improvement of carotenoid, chlorophyll, protein content, nitrate reductase, and NAD-Glutamate Dehydrogenase activities in plants’ leaves compared to the control [25]. In addition, it has been shown that glucose, sucrose, or trehalose-6-phosphate can regulate a number of growth and metabolic processes, acting independently of the basal functions; they can also act as signalling molecules [24]. Additionally, protein hydrolysates might enhance biological activity in crop growth and development [26]. Plants treated with the protein hydrolysate showed better water status and pollen viability, which also resulted in a higher yield under drought stress compared to untreated plants [27]. The treatment with the protein hydrolysate had an effect on antioxidant contents and activity in leaves and fruits depending on the level of irrigation provided [27]. Furthermore, microalgae contain a wide variety of bioactive compounds; extraction of these components from the biomass may be necessary. The cell wall of microalgae is a multi-layered structure acting as a physical barrier to solvent input; it is the main thing responsible for the isolation of the intracellular core of the cell, in which some compounds are concentrated [3]. In this assay, the flower number increased compared to the control by 56% in treatments with ultrasound, and 59% in treatments with ultrasound, enzymatic hydrolysis, and autoclaving. Tejada et al. [21] observed a similar increase (51%), when silicon and *Arthrospira* were jointly applied to the plant. Venture et al. [3] observed that, compared to conventional extraction, ultrasound-assisted extraction is a more efficient and rapid extraction method due to the strong disruption of the cell wall that is achieved. To facilitate the release of important biomolecules from microalgae, the use of ultrasound has gained tremendous interest as an alternative to traditional methods [28]. Ultrasound-assisted extraction is a suitable alternative for the extraction of several bioproducts, as it confers several advantages, such as short treatment time and solvent consumption, high efficiency in cell disruption, high extraction yields, suitability to extract thermolabile compounds, as well as being an inexpensive extraction method [29]. Furthermore, ultrasonic-assisted extraction has no effect on the target compounds’ chemical structure or biological properties [29]. It is difficult to establish a direct relationship between one active compound and a stimulating effect because the availability of the bioactive compound is not always well known, and interaction of the components can take place. *C. vulgaris* has a high concentration of proteins, carbohydrate lipids, and pigments [10]. An efficient lipid extraction technique is necessary since the thick cell wall of several microalgae species blocks the release of intracellular lipids [9]. Normally, lipids are extracted using a non-water miscible organic solvent, but an ultrasound assisted extraction technique has been used [9]. Plant lipids are diverse, essential for cells [30], and play a role in flower development. Very long-chain fatty acids are involved as membrane constituents and signalling molecules in sphingolipids and phospholipids are necessary for the production of cuticular waxes. Phospholipids and complex sphingolipids have, collectively, profound effects on embryo, leaf, root, and flower development [25]. Further application of ultrasound successfully disintegrated microalgal cells and resulted in increased concentrations of protein and carbohydrates [31], carotenoids [32] and chlorophylls [33]. Chlorophyll affects the rate of photosynthesis and carotenoids participate in various biological processes in plants, such as photosynthesis, photomorphogenesis, photoprotection, and development [34]. Hajna-Jafari et al. [35] found that foliar treatment with *C. vulgaris* S45 significantly increased carotenoid and chlorophyll content compared to the control. To extract bioactive compounds from eukaryotic microalgae for plant biostimulant purposes, the US-treatment (or any other method damaging the plasma membrane) of microalgae cells is, or seems to be, suitable.

Additionally, the extraction of bioactive compounds involves complex mechanisms, and can be achieved by combining several techniques. In this sense, combinations of methods can sometimes produce satisfactory results, whereas one method alone fails [36]. Methods with different specificities may be combined synergistically [37]. In this assay, the application of the microalgae improved plant performance, but treating the biomass with the different methods (M-USHY and M-USHYAU) did not accrue any benefits compared to the ultrasound treatment. Contrary to the findings in this study, Salinas et al. [38] reported that ultrasonic-assisted extraction can perform well when coupled with enzymatic treatment. On the other hand, microalgae foliar application with M-USHY reduced leaves and flowers as well as stem and petiole dry weight compared to M and M-USHYAU. The use of ultrasound and enzymatic hydrolyses without autoclaving produces negative effects. These treatments may make substances available that are harmful to the plant and that can be neutralised only after autoclaving. Hopkins [36] reported that there are several potential problems with using enzymatic lysis, one of them being that the product to be recovered from the cell might be destroyed or modified during lysis. Furthermore, after cell disruption, the compounds that were protected in the cell will be exposed to oxidation, which may result in the decomposition of active compounds [39]. However, Middelberg [37] found that enzymatic treatment prior to mechanical disruption provides clear benefits due to the high ratio of cell walls in microalgae that might require an enzymatic treatment for enhanced cell rupture [26]. Chiaiese et al. [10] reported that the significant advantages of the enzymatic pre-treatment methods over the chemical ones have been mainly attributed to the gentler cell-wall disruption, which does not involve chemical and/or physical treatment.

Furthermore, contrary to the results of other authors, nutrient contents were not positively affected by applying the microalgae to the crop. Kusvuran [40] reported an increase in nutrient uptake with the application of microalgae (*C. vulgaris* Beijerinck). In addition, the extraction method applied to the microalgae does not affect the nutrient concentration in the leaves of the *Pelargonium* × *hortorum*.

## 5. Conclusions

Based on the results obtained, ultrasound treatment promotes early flowering and increases the number of inflorescences compared to the control. The US-treated *C. vulgaris* biomass is the most suitable for *Pelargonium* treatment. All the *C. vulgaris* applications stimulated lateral shoot growth and increased stem and plant diameter. The extraction method applied to the microalgae biomass does not affect nutrient leaf content. Further research is required in this area to fully utilize the potential of microalgae biomass.

## Figures and Tables

**Figure 1 plants-11-02308-f001:**
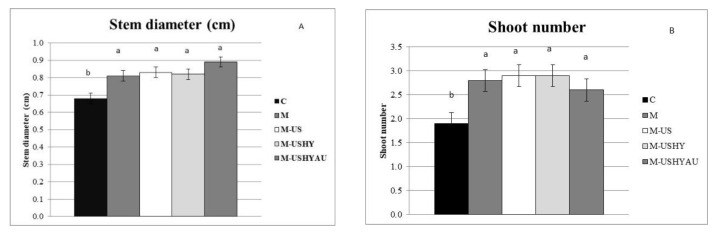
Stem diameter (cm) (**A**) and shoot number (**B**). Data are means ± standard error. Values followed by different letters are significantly different (*p* ˂ 0.05). The values are the means of data from 12 plants for each replicate.

**Figure 2 plants-11-02308-f002:**
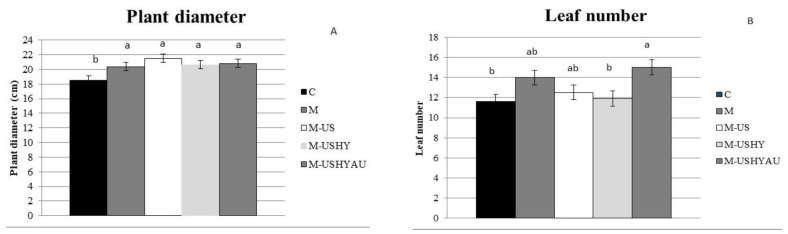
Plant diameter (cm) (**A**) and number of leaves (**B**). Data are means ± standard error. Values followed by different letters are significantly different (*p* ˂ 0.05). The values are the means of data from 12 plants for each replicate.

**Figure 3 plants-11-02308-f003:**
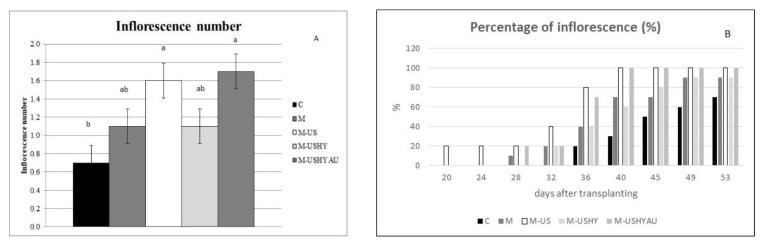
Inflorescence number per plant (**A**) and percentage of inflorescence (**B**). Data are means ± standard error. Values followed by different letters are significantly different (*p* ˂ 0.05). The values are the means of data from 12 plants for each replicate.

**Figure 4 plants-11-02308-f004:**
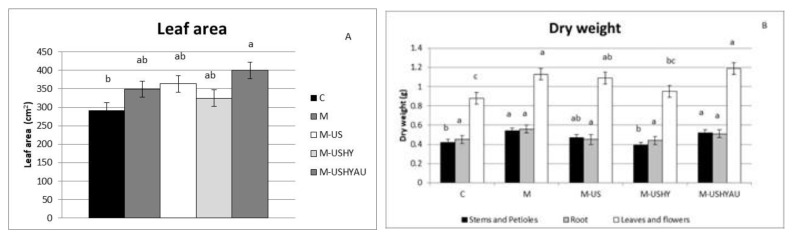
Leaf area (cm^2^) (**A**) and dry weight (**B**). Data are means ± standard error. Values followed by different letters are significantly different (*p* ˂ 0.05). The values are the means of data from 12 plants for each replicate.

**Figure 5 plants-11-02308-f005:**
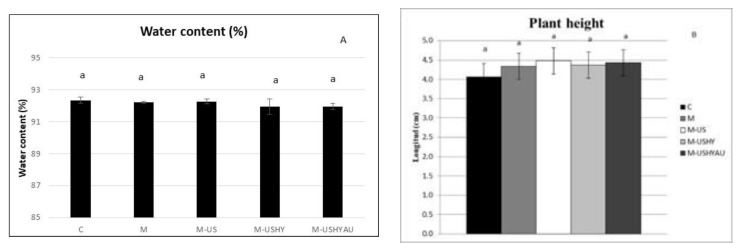
Water content (%) (**A**) and plant height (cm) (**B**). Data are means± standard error. Values followed by different letters are significantly different (*p* ˂ 0.05). The values are the means of data from 12 plants for each replicate.

**Figure 6 plants-11-02308-f006:**
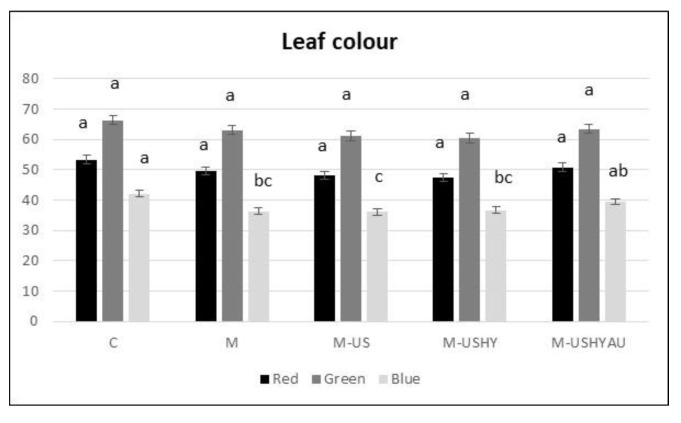
Leaf colour (red, green, and blue). Data are means ± standard error. Values followed by different letters are significantly different (*p* ˂ 0.05). The values are the means of data from 12 plants for each replica.

**Figure 7 plants-11-02308-f007:**
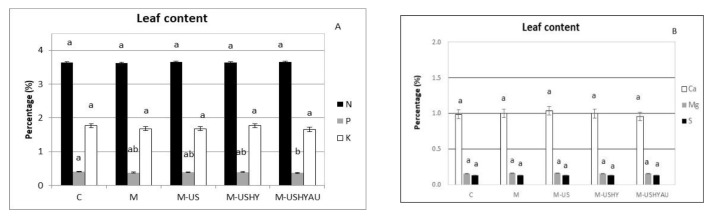
Leaf N, P, and K content (%) (**A**) and leaf Ca, Mg, and S content (**B**). Data are means ± standard error. Values followed by different letters are significantly different (*p* ˂ 0.05). The values are the means of data from 12 plants for each replica.

**Table 1 plants-11-02308-t001:** Hormone content of the untreated microalga (ng g^−1^).

*C. vulgaris*		
Ethylene	ACC	11.19
Cytokinins	Trans-Zeatin	813.65
	Isopentenyl-adenine	3816.28
Gibberellins	GA1	7.28
	GA3	0.01
	GA4	26.62
Auxins	Indoleacetic acid	2.53
Other hormones	ABA	2.17
	Salicylic acid	408.86

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
