# Peer review of "Effect of the Application of Hydrolysate of Chlorella vulgaris Extracted by Different Techniques on the Growth of Pelargonium × hortorum"

_plants, 2022, doi:10.3390/plants11172308_

Round 1
Reviewer 1 Report (Previous Reviewer 1)
General comments:
The paper was revised with limited attention. It needs major revision again.
Other comments to the paper:
- Line 27: biostimulating instead of “biostimulanting”
- Line 79: contains instead of “content”
- Lines 81-84: this establishment originates from Faheed and Abd-El Fattah, 2008, and relates to lettuce plants. Therefore, the correct sentence: “Faheed and Abd-El Fattah (2008) reported that improvement of lettuce plant growth treated with C. vulgaris is associated to the stimulation of carotenoid and chlorophyll pigment biosynthesis, which may have improved the photosynthetic activity.”
- Line 94: microalga instead of “microalgae”
- Lines 99-100: it is not possible to obtain a concentration of “250 g L-1(dry weight)”. Please recalculate this value and consider that: (1) biomass concentration of the Chlorella suspension at the harvest day; (2) dry matter content of the algae cells is between 10 and 20%. Was the harvested wet biomass dried, freeze-dried, or used in wet form?
- Lines 102-110: How many replicates and plants were used in each treatment?
- Lines 111-112: Please add the dry matter concentration of the biomass suspension, which was used processed?
- Lines 118-121: the description of the method is not clear enough, please improve it.
- Please introduce the correct figure: Fig. 2A (plant diameter) is the same as Fig. 1A. (stem diameter).
- Line 229: The number of inflorescence caused by the biomass “treated with ultrasound, enzymatic hydrolysis and autoclaving” is much higher than 60% of the control – higher than double of the control, please re-calculate it.
- Line 236: M-US instead of “M-AS”.
- Fig. 3A and 3B show contradictions. Let’s suppose that Fig. 3A shows the inflorescence number at the end of the experiment (day 53). In this case Fig. 3B values on day 53 are not correct.
- Lines 250-257: Please compare the values written in the text and shown on Fig. 4B, because there are differences in the values, which should be the same. Please also check the significant differences mentioned in the text.
- Line 284: “Rhododendron indicum” in italic.
- Lines 311-313: This establishment is not correct, if it is about the results of the present paper: “In this assay, the application of the microalgae improved plant performance but treating the biomass with the different method (ultrasound and/or enzymatic hydrolysis and/or autoclaving) did not give any benefits compared with the raw material.”
- Lines 338-341: This establishment is either not correct or not properly describe if the “raw material” is the harvested Chlorella suspension (M).

Author Response
We would like to thank the reviewer for careful and for the thoughtful comments and constructive suggestions, which help to improve the quality of this manuscript. The text has been revised as suggested:
Review 1
Pelargonium Chlorella hydrolysate paper 1827362 – Plants General comments: The paper was revised with limited attention. It needs major revision again.
Other comments to the paper:
- Line 27: biostimulating instead of “biostimulanting” The correction has been made.
-Line 79: contains instead of “content” The correction has been made.
- Lines 81-84: this establishment originates from Faheed and Abd-El Fattah, 2008, and relates to lettuce plants. Therefore, the correct sentence: “Faheed and Abd-El Fattah (2008) reported that improvement of lettuce plant growth treated with C. vulgaris is associated to the stimulation of carotenoid and chlorophyll pigment biosynthesis, which may have improved the photosynthetic activity.” The correction has been made.
- Line 94: microalga instead of “microalgae” The correction has been made.
- Lines 99-100: it is not possible to obtain a concentration of “250 g L-1 (dry weight)”. Please recalculate this value and consider that: (1) biomass concentration of the Chlorella suspension at the harvest day; (2) dry matter content of the algae cells is between 10 and 20%. Was the harvested wet biomass dried, freeze-dried, or used in wet form? The correction has been made. The original C. vulgaris culture got a concentration of 2 g/L, it was centrifuged with Alfa laval centrifuge concentrating the biomass up to 200 g/L (20%DW) this final concentration has been reported in other publications (M. Al Hattab, A. Ghaly, A. Hammouda, Microalgae harvesting methods for industrial production of biodiesel: critical review and comparative analysis, J. Fundam. Renew. Energy Appl. 05 (2015), https://doi.org/10.4172/2090-4541.1000154). And it was diluted to the desirable concentration, 100 g/L (10%DW).
- Lines 102-110: How many replicates and plants were used in each treatment? The correction has been made. There were 12 replicates per treatment.
- Lines 111-112: Please add the dry matter concentration of the biomass suspension, which was used processed? The correction has been made. To achieve to desired products for the application of the treatment, the biomass at 100 g/L was processed as follows:
- Lines 118-121: the description of the method is not clear enough, please improve it. The correction has been made. Enzymatic protein hydrolysis of the ruptured biomass was carried out using jacketed reactors, stirred via an impeller; the temperature was 40 ºC and the pH 8, which was established by the addition of sodium hydroxide (NaOH) 1N. For hydrolysis, the enzymes Alcalase 2.5L and Flavourzime 1000L (Novozyme) were used, at a dose of 7% v/w of protein content. The process lasted 4 hours.
- Please introduce the correct figure: Fig. 2A (plant diameter) is the same as Fig. 1A. (stem diameter). The correction has been made.
- Line 229: The number of inflorescence caused by the biomass “treated with ultrasound, enzymatic hydrolysis and autoclaving” is much higher than 60% of the control – higher than double of the control, please re-calculate it. The increase was estimated as (MUSHYAU Inflorescence number -Control Inflorescence number) / MUSHYAU Inflorescence number, meaning (1.7-0.7)/1.7= 60 %
- Line 236: M-US instead of “M-AS”. The correction has been made.
- Fig. 3A and 3B show contradictions. Let’s suppose that Fig. 3A shows the inflorescence number at the end of the experiment (day 53). Figure 3B present number of inflorescences per plants, one plant can have more than one, but figure 3B present the percentage of plant with inflorescence, 100 % mean all the plant has inflorescence (one inflorescence or more).
- Lines 250-257: Please compare the values written in the text and shown on Fig. 4B, because there are differences in the values, which should be the same. Please also check the significant differences mentioned in the text. The correction has been made.
- Line 284: “Rhododendron indicum” in italic. The correction has been made.
- Lines 311-313: This establishment is not correct, if it is about the results of the present paper: “In this assay, the application of the microalgae improved plant performance but treating the biomass with the different method (ultrasound and/or enzymatic hydrolysis and/or autoclaving) did not give any benefits compared with the raw material.” I have eliminated the sentence.
- Lines 338-341: This establishment is either not correct or not properly describe if the “raw material” is the harvested Chlorella suspension (M). I change as untreated microalga and I have removed this sentence.
Reviewer 2 Report (New Reviewer)
Before getting into the in-depth analysis of this article for publication, the following clarifications are needed from the author’s side.
1. What is the novelty of this study? as there are many articles on the use of chlorella hydrolysate for plant growth?
2. Usage of algal extract for increasing the shoot propagation/plant growth by foliar spray or other modes of application is experimented by many.
3. What are the compounds in the algal hydrolysate?. Or what are the compositional differences in the hydrolysates extracted by different methods? it is critical.
4. Discussion should be improved. New findings of this study compared to the previously published reports should be included.
5. What is the molecular or cellular mechanism by which chlorella hydrolysate increases plant growth?
Author Response
We would like to thank the reviewer for careful and for the thoughtful comments and constructive suggestions, which help to improve the quality of this manuscript. The text has been revised as suggested:
Review 2
Before getting into the in-depth analysis of this article for publication, the following clarifications are needed from the author’s side.
1.What is the novelty of this study? as there are many articles on the use of chlorella hydrolysate for plant growth? I have included:
In particularly, Chlorella have been extensively studied to obtain biomass or to extract bioactive compounds with potential applications in functional food supplements, as well as nutraceuticals, cosmetics, and pharmaceuticals [7b]. While cell disruption methods have been optimized for lipid and protein extraction, there are limited studies for other bioactive compounds [26a], especially the ones with an interest in agriculture.
2.Usage of algal extract for increasing the shoot propagation/plant growth by foliar spray or other modes of application is experimented by many. I have included:
While cell disruption methods have been optimized for lipid and protein extraction, there are limited studies for other bioactive compounds [26a], especially the ones with an interest in agriculture.
3.What are the compounds in the algal hydrolysate?. Or what are the compositional differences in the hydrolysates extracted by different methods? it is critical. I have included:
The proteins, lipids, and carbohydrates content in untreated C. vulgaris was 42.68±1.07, 3.80±0.40 and 46.48±0.80 % respectively. The hormone content of the microalga was ACC 11.19 ng g-1, trans-Zeatin 813.65 ng g-1, Isopentenyl-adenine 3816.28 ng g-1, GA1 7.28 ng g-1, GA4 26.62 ng g-1, Indoleacetic acid 2.53 ng g-1, ABA 2.17 ng g-1, Salicylic acid 408.86 ng g-1.
- Discussion should be improved. New findings of this study compared to the previously published reports should be included. The correction has been made. I have included 8 references in discussion section, related to ultrasound extraction
- What is the molecular or cellular mechanism by which chlorella hydrolysate increases plant growth? The correction has been made. I have included the possible molecular mechanism by which chlorella increase plant growth in discussion section:
It is difficult to establish a direct relationship between one active compound and a stimulating effect, due to the fact that the availability of the bioactive compound is not always well known, and interaction of the components can take place. C. vulgaris are rich sources of proteins, lipids, carbohydrates and pigments (Panahi et al., 2019). Application of ultrasound successfully disintegrated microalgal cells and resulted in increased concentrations of proteins and carbohydrates (Keris-Sen et al., 2014), carotenoids (Deenu et al., 2013) and chlorophylls (Parniakov et al., 2015)............
Round 2
Reviewer 2 Report (New Reviewer)
The authors have not responded well to the comments/suggestions given. Revise the manuscript carefully to avoid rejection.
1. Comment no: 1 is not responded well. What is the key novelty!!!
2. Comment no: 1 : not responded well. chlorella bioactive compounds are widely studied.
3. Comments no: 3: I could not find a meaningful discussion by citing previously published papers. Revise again.
4. In the Introduction: starting from” Microalgae are unicellular microorganisms 47 which thrive in saltwater, freshwater environments [4] and wastewater………………. functional food supplements, as well as nutraceuticals, cosmetics, 90 and pharmaceuticals”- this part is incomplete, which needs to be revised with more information on introduction to algae, biochemical components, and products in algae and their application with recent literature support. Refer to these papers and revise to increase the archival worth of the manuscript. a. https://doi.org/10.1016/j.bcab.2018.12.007, b. https://doi.org/10.1016/j.bcab.2019.01.017, c. https://doi.org/10.1016/j.jclepro.2019.04.287.
5. All algal cells release protein, carbohydrates, lipids, and pigments upon cell disruption. It is known and understood. What is the actual mechanism by which algal metabolites promote plant growth in this study?.
Author Response
We would like to thank the reviewer for careful thoughtful comments and constructive suggestions, which help to improve the quality of this manuscript. I hope I have responded well to the comments of the reviewer. The text has been revised as suggested:
The authors have not responded well to the comments/suggestions given. Revise the manuscript carefully to avoid rejection.
1.Comment no: 1 is not responded well. What is the key novelty!!!
2.Comment no: 1 : not responded well. chlorella bioactive compounds are widely studied. I have included following paragraph: .
C. vulgaris is a source of high value bioactive compounds. Efficient extraction technique is necessary to obtain maximal bioactive compounds content from microalgae. Recent efforts on extraction and purification of some components from microalgae have been growing but more systematic studies are needed to identify economical methods to produce maximum bioactive compounds from marine sources (Ramesh Kumar et al., 2019 Mathimani and Pugazhendhi, 2019). .Additionally, the bioactive compounds of the microalgae can exert synergistic effects, therefore in vivo studies are needed to establish the most appropriate extraction method (Sudhakar et al., 2019).
3.Comments no: 3: I could not find a meaningful discussion by citing previously published papers. Revise again. I have included the references recommended by the author and five more.
Efficient lipid extraction technique is necessary, since the thick cell wall of several microalgae species, blocks the release of intracellular lipids (Ramesh-Kumer et al, 2019). The extraction of lipids is usually done using a non-water miscible organic solvent, however ultrasound assisted extraction technique has been also employed (Ramesh-Kumer et al, 2019). Plants lipids are diverse, essential for cells (Kim , 2020) and play a role in flower development. Very long chain fatty acids are involved as membrane constituents and signaling molecules in sphingolipids and phospholipids and are necessary for the production of cuticular waxes phospholipids and complex sphingolipids have, collectively, profound effects on embryo, leaf, root and flower development (Rahidi et al., 2020).
4.In the Introduction: starting from” Microalgae are unicellular microorganisms 47 which thrive in saltwater, freshwater environments [4] and wastewater………………. functional food supplements, as well as nutraceuticals, cosmetics, 90 and pharmaceuticals”- this part is incomplete, which needs to be revised with more information on introduction to algae, biochemical components, and products in algae and their application with recent literature support. Refer to these papers and revise to increase the archival worth of the manuscript. a. https://doi.org/10.1016/j.bcab.2018.12.007, b. https://doi.org/10.1016/j.bcab.2019.01.017, c. https://doi.org/10.1016/j.jclepro.2019.04.287.
I have included the references recommended by the reviewer in introduction and discussion section.
Ramesh Kumar, Garlapati Deviram, Thangavel Mathimani, Pham Anh Duc, Arivalagan Pugazhendhi,Microalgae as rich source of polyunsaturated fatty acids, Biocatalysis and Agricultural Biotechnology, 17, 2019, 583-588,https://doi.org/10.1016/j.bcab.2019.01.017.
Thangavel Mathimani, Arivalagan Pugazhendhi, Utilization of algae for biofuel, bio-products and bio-remediation, Biocatalysis and Agricultural Biotechnology,17,2019,326-330, https://doi.org/10.1016/j.bcab.2018.12.007.
M.P. Sudhakar, B. Ramesh Kumar, Thangavel Mathimani, Kulanthaiyesu Arunkumar, A review on bioenergy and bioactive compounds from microalgae and macroalgae-sustainable energy perspective, Journal of Cleaner Production, 228, 2019, 1320-1333, https://doi.org/10.1016/j.jclepro.2019.04.287.
- All algal cells release protein, carbohydrates, lipids, and pigments upon cell disruption. It is known and understood. What is the actual mechanism by which algal metabolites promote plant growth in this study? The correction has been made. I have included following paragraphs:
Protein hydrolysates might enhance biological activity in crop growth and development [24]. Plants treated with the protein hydrolysate showed a better water status and pollen viability, which also resulted in higher yield under drought stress compared to untreated plants (Francesca et al., 2021). The treatment with the protein hydrolysate had also an effect on antioxidant contents and activity in leaves and fruits depending on the level of irrigation provided (Francesca et al. 2021 ).
Carbohydrates play an important role in the life of plants: they are structural and storage substances, respiratory substrates, and intermediate metabolites of many biochemical processes (Ciereszko, 2018 ). The application of microalgae polysaccharides showed an improvement of carotenoid, chlorophyll and proteins content, and Nitrate Reductase, NAD-Glutamate Dehydrogenase activities in plants leaves compared to control (Rachidi et al., 2020). Additionally, it has been shown that glucose, sucrose, or trehalose-6-phosphate can regulate a number of growth and metabolic processes, acting independently of the basal functions; they can also act as signaling molecules (Ciereszko, 2018) .
Plants lipids are diverse, essential for cells (Kim, 2020) and play a role in flower development. Very long chain fatty acids are involved as membrane constituents and signaling molecules in sphingolipids and phospholipids and are necessary for the production of cuticular waxes phospholipids and complex sphingolipids have, collectively, profound effects on embryo, leaf, root and flower development (Rahidi et al., 2020). Efficient lipid extraction technique is necessary, since the thick cell wall of several microalgae species, blocks the release of intracellular lipids (Ramesh-Kumer et al, 2019).
Round 3
Reviewer 2 Report (New Reviewer)
ACCEPT
This manuscript is a resubmission of an earlier submission. The following is a list of the peer review reports and author responses from that submission.
Round 1
Reviewer 1 Report
The report is introduced as attached file.

Reviewer 2 Report
The manuscript by Bayona-Morcillo et al. has some serious deficits regarding the design of the study therefore it cannot be accepted for publication. First, the main scope of this study is to determine the most efficient extraction method to release the active components of Chlorella vulgaris (as also claimed in the abstract of the manuscript), as well as to evaluate the effect of adding these microalgae samples after extraction to the growth of Pelargonium x hortorum. However, no assessment of the extraction process prior to the plant experiments has been conducted, for example following a simple process such as the determination of a bioactive compound that is characteristic for C. vulgaris extracts. In this way, the evaluation of the extraction process is done indirectly through plant growth experiments, which is a labor work requiring time and effort. Preliminary assessment of the extraction process might be crucial for the selection of the appropriate method and/or conditions in order to maximize the extracted compounds.
Moreover, it is not clear why authors used only protein-degrading enzymes, especially when the cell wall of C. vulgaris is consisted of cellulose (as also mentioned in the manuscript). Perhaps, this is the reason why no improvement was observed when enzymatic hydrolysis was combined with ultrasound treatment. As far as concerned the use of autoclave, the extracts contain several compounds such as fatty acids or phytohormones; do authors speculate that these compounds will not be destroyed in such high temperatures?
Some other comments that could generally help the authors:
The experimental part should be re-written, by describing first the growth of microalgae, the extraction process and finally the plant experiments. Moreover, regarding the enzyme treatment, the loading should be expressed in mg of protein/g of substrate and not in volume units!
In the Discussion section, a numerical comparison of the results of the study with other works in the literature should be also included, so as to highlight the contribution of this work. The novelty aspects of the work should be also written clearly in the manuscript.
The full name of microorganisms (Chlorella vulgaris) should be given only the first time that is mentioned in the manuscript and the short name C. vulgaris should be used thereafter.